# Genome-Wide Identification and Analysis of CC-NBS-LRR Family in Response to Downy Mildew and Black Rot in Chinese Cabbage

**DOI:** 10.3390/ijms22084266

**Published:** 2021-04-20

**Authors:** Yan Liu, Dalong Li, Na Yang, Xiaolong Zhu, Kexin Han, Ran Gu, Junyu Bai, Aoxue Wang, Yaowei Zhang

**Affiliations:** 1Key Laboratory of Biology and Genetic Improvement of Horticulture Crops (Northeast Region), Ministry of Agriculture and Rural Affairs, Northeast Agricultural University, Harbin 150000, China; liuyanworld@126.com (Y.L.); lilong00000@163.com (D.L.); m13789700617@163.com (N.Y.); zxl1249991744@163.com (X.Z.); a370851240@163.com (K.H.); guran@cau.edu.cn (R.G.); baijunyu11@126.com (J.B.); 2College of Horticulture, Northeast Agricultural University, Harbin 150030, China

**Keywords:** Chinese cabbage, CC–NBS–LRR, downy mildew, black rot

## Abstract

The nucleotide-binding site–leucine-rich repeat (NBS–LRR) gene family is the largest group of plant disease resistance (R) genes widespread in response to viruses, bacteria, and fungi usually involved in effector triggered immunity (ETI). Forty members of the Chinese cabbage CC type NBS–LRR family were investigated in this study. Gene and protein characteristics, such as distributed locations on chromosomes and gene structures, were explored through comprehensive analysis. CC–NBS–LRR proteins were classified according to their conserved domains, and the phylogenetic relationships of CC–NBS–LRR proteins in *Brassica rapa*, *Arabidopsis thaliana*, and *Oryza sativa* were compared. Moreover, the roles of *BrCC–NBS–LRR* genes involved in pathogenesis-related defense were studied and analyzed. First, the expression profiles of *BrCC–NBS–LRR* genes were detected by inoculating with downy mildew and black rot pathogens. Second, sensitive and resistant Chinese cabbage inbred lines were screened by downy mildew and black rot. Finally, the differential expression levels of *BrCC–NBS–LRR* genes were monitored at 0, 1, 3, 6, 12 and 24 h for short and 0, 3, 5, 7, 10 and 14 days for long inoculation time. Our study provides information on *BrCC–NBS–LRR* genes for the investigation of the functions and mechanisms of *CC-NBS-LRR* genes in Chinese cabbage.

## 1. Introduction

Plants have developed specific mechanisms to protect themselves from abiotic or biotic stress [1,2]. Plant pathogen responses are rapidly activated when plants are attacked by pathogens, such as bacteria, fungi, oomycetes, viruses, nematodes, and insects, protecting plants from further harm [3]. Plant disease resistance (R) genes are involved in defense against pathogens, are triggered by pathogen signaling, and can target specific pathogens [4]. Most R genes encode a kind of nucleotide-binding site–leucine-rich repeat (NBS–LRR) protein. The NBS domain belongs to the NB-ARC domain and contains three conserved motifs, namely, P-loop, kinase-2, which coordinates divalent metal ions, and kinase-3a-binding nucleotide, which is a critical region for nucleotide binding in ATP/GTPs and plays a role in defense signal transmission [5]. Most LRR domains with 20–30 amino acid residues consist of two segments: a highly conserved segment (HCS) and variable segment (VS). However, the domains are usually 23–25 residues long, and a specific VS part in a plant LRR domain is involved in plant pathogen immune reaction and protein-to-protein interactions [6]. The NBS–LRR gene family is the largest class of R genes, which play multiple roles in direct or indirect host–pathogen recognition and downstream signaling transduction [7,8].

NBS–LRR proteins are divided into two types according to their conservative functional domains. One contains a toll/interleukin 1 receptor (TIR) domain at the N terminal and the other is the non-TIR(CC–NBS–LRR type), which assembles a coiled-coil (CC) domain at the N terminal instead of a TIR domain. Other domains in the N-terminal of CC–NBS–LRR replace or coexist with CC domains, particularly zinc fingers or RPW8 domains [9,10]. CC–NBS–LRR genes are found in monocotyledons and dicotyledons, and *toll/interleukin-1 receptor-NBS-LRR* (*TNL*) genes are rare in monocotyledons [9], revealing that CC–NBS–LRR is widely present in plants. Furthermore, approximately 80% of reported R genes encode the NBS–LRR domain, and more than 50 NBS genes have been proved in response to disease resistance [11]. The R gene Pi-ta belonging to the NBS-LRR type played a direct role in Magnaporthe grisea by interaction with the effector AVR-Pita in rice [12]. Additionally, the RRS1 protein of *Arabidopsis thaliana* interacted directly with PopP2, which was the pathogen protein of bacterial wilt [13]. RPS2 and RPM1 resistance genes from *Arabidopsis* were in response to *Pseudomonas syringae* by indirect interaction with AvrRpm1 and AvrB [14,15]. The ectopic overexpression of *Arabidopsis RPW8* enhanced the resistant ability to powdery mildew in grapevine [16].

Chinese cabbage is a popular vegetable worldwide, especially in Asia. The leaf heading, which forms at the late heading stage, is the main edible part and commodious organ and is directly associated with the yield and quality of Chinese cabbage [17]. Downy mildew (DM) and black rot (BR) are the two common diseases affecting the heading leaves of Chinese cabbage in China and decreasing quality and yield [18]. Caused by *Hyaloperonospora parasitica*, DM is one of the three major diseases of Chinese cabbage and even the entire cruciferous family. BR is a bacterial disease caused by *Xanthomonas campestris* pv. *campestris dowson*, which infects many *Brassica* species, such as cabbage (*Brassica oleracea* var), Chinese cabbage (*Brassica pekinensis*), and oil seed rape (*Brassica* campestris) [19,20]. Thus, screening germplasm and identifying tolerant lines of Chinese cabbage will be helpful.

A total of 92 NBS-encoding resistance genes of *Brassica* rapa were identified without a reference genome [21], and the expression levels of 90 TIR–NBS–LRR genes against TuMV have been analyzed in *Brassica rapa* ssp. *Pekinensis* [22]. CC–NBS–LRR proteins in Chinese cabbage have not been explored, especially their roles in defense against bacterial and fungal infection. In our study, the 40 *CC–NBS–LRR* genes of Chinese cabbage were identified and analyzed in a genome-wide range. The basic information of the BrCC–NBS–LRR family was listed, including Genome position, exon No., the length of CDS and Protein, etc. A phylogenetic tree of *CC–NBS–LRR* genes in *Brassica rapa*, *Arabidopsis thaliana*, and *Oryza sativa* was built to elucidate the evolution. The expression profiles of *BrCC–NBS–LRR* genes in different tissues and in response to diseases were detected and analyzed, showing that *BrCC–NBS–LRR* genes were enriched in leaf, and their responses were quite different due to the different types of pathogen and time. Furthermore, the sensitive and insensitive inbred lines were obtained and inoculated with *Hyaloperonospora parasitica* and *Xanthomonas campestris* to explore the predicting functions of *BrCC–NBS–LRR* genes. These results suggest that they play an opposite or consistent function to diseases. Our study provides information on *BrCC–NBS–LRR* genes. This information is useful to further investigate the gene functions and mechanisms of *CC–NBS–LRR* genes in Chinese cabbage.

## 2. Results

### 2.1. Identifcation of the CC–NBS–LRR Family in Brassica rapa

The CC–NBS–LRR family usually encodes two typical domains, including the CC domain at the C terminal and the NBS–LRR domain at the N terminal. All *CC–NBS–LRR* family genes were searched using the *Brassica rapa* genome on Brassica Database (BRAD). A total of 40 CC–NBS–LRR family candidates were found, and their protein sequences were verified using online tools for protein functional domains. The tools were PfamScan (https://www.ebi.ac.uk/Tools/pfa/pfamscan/, accessed on 27 February 2021. EMBL-EBI 2013), InterPro (a protein sequence analysis and classifications website, http://www.ebi.ac.uk/interpro/, accessed on 27 February 2021. 2021), and ScanProsite (https://prosite.expasy.org/scanprosite/, accessed on 27 February 2021. 2018). Finally, 40 genes were screened, all of which have intact CC, NBS, and LRR domains. The *CC–NBS–LRR* family genes of Chinese cabbage were named according to their genes that were ID-searched on BRAD. Blast searches for the homologous genes of *BrCC–NBS–LRR* in *Arabidopsis thaliana* were conducted using TAIR (https://www.arabidopsis.org/, 31 March 2005), and 20 genes were homologous to CC–NBS–LRR members shown in Appendix A. In our results, the main genetic characteristics of *CC–NBS–LRR* genes in *Brassica rapa* are summarized and shown in Table 1, including the gene ID chromosome location, the number of exons, protein length, molecular weights, and isoelectric point (PI). In all BrCC–NBS–LRR genes, exon length was between 1992 and 4842 bp, and the amount varied from 1 to 12. As for the proteins, the length ranged from 666 aa to 1613 aa. The isoelectric points between 5.42 and 8.98 are shown in Table 1.

### 2.2. Location and Distribution of CC–NBS–LRR Family Genes on Brassica rapa Chromosome

The physical chromosomal locations of *BrCC–NBS–LRR* family genes can be found on BRAD (http://brassicadb.org/brad/index.php, accessed on 27 February 2021), and are drawn and shown in Figure 1. Apart from A04 and A07, the BrCC–NBS–LRR family members showed asymmetric distribution on each chromosome of *Brassica rapa* from A1 to A10. The majority of *BrCC–NBS–LRR* genes were located on A09 with 12 members (30%), followed by A06 with 8 (15%) BrCC–NBS–LRR members. However, only one member was located on A02. Two members were located on A05 and A10, and four, five, and six members were located on A3, A01, and A08, respectively. A total of 24 *CC–NBS–LRR* family genes were found in *Arabidopsis thaliana*, 16 (67%) of which were located on Chr.1 and eight were located on Chr.5 (Appendix A). However, 40 members of *BrCC–NBS–LRR* have uneven distribution on the eight chromosomes of Chinese cabbage because of chromosome expansion. Duplication analysis showed that only four pairs of genes had two copies, and two genes had three copies, which were distributed on different chromosomes. This result suggested that duplicate events in Chinese cabbage were caused by segment duplications without tandem.

### 2.3. Phylogenetic Analysis of the CC–NBS–LRR Family in Brassica rapa, Arabidopsis thaliana, and Oryza sativa

To further analyze the phylogenetic relationship of *CC–NBS–LRR* family genes among Chinese cabbage, *Arabidopsis thaliana*, and rice, we aligned a series of multiple protein sequences and built a phylogenetic tree with the neighbor-joining method. MEGA7.0 was used, and 40 CC–NBS–LRR members of Brassica rapa, 24 AtCC–NBS–LRR genes, and eight candidates belonging to rice were included (Figure 2, Appendix A). The CC–NBS–LRR proteins were divided into nine subgroups according to their protein sequence features, namely, Groups A to G. The largest subgroup was Group A with 14 members, including 12 BrCC–NBS–LRR and two AtCC–NBS–LRR proteins. Groups E and H had four proteins each and thus had the smallest subgroups in terms of number. All the proteins in Group E were Chinese cabbage CC–NBS–LRR proteins, and two proteins in Group H were Chinese cabbage proteins and two were *Arabidopsis thaliana*, showing that Bra013134 shared high homology with AT1G58400 and Bra035424 shared high homology with At1G59620. Two types of CC–NBS–LRR proteins were found in Group B, including four AtCC–NBS–LRR members and seven BrCC–NBS–LRR members, with a total of 11 members. Similar to Group B, Group C contained four AtCC–NBS–LRR and five BrCC–NBS–LRR proteins. Seven CC–NBS–LRR proteins, including two *Brassica rapa* and five *Oryza sativa* members, were found. Groups F and G had three types of CC–NBS–LRR proteins, such as *Arabidopsis thaliana*, *Oryza sativa*, and *Brassica rapa* proteins. In Group F, two AtCC–NBS–LRR, two OsCC–NBS–LRR, and four BrCC–NBS–LRR proteins were found, whereas Group G had one *Oryza sativa*, two *Arabidopsis thaliana*, and three *Brassica rapa* members. These results showed that the phylogenetic tree of all the CC–NBS–LRR members is inclined to species differences because of the loose functional domain structures of the CC domain, NBS, and LRRs.

### 2.4. Gene Structures and Protein Function Domains of CC–NBS–LRR Family in Chinese Cabbage

The gene structures and protein function domains of the BrCC–NBS–LRR family were analyzed using a phylogenetic tree built by aligning all the protein sequences of BrCC–NBS–LRR proteins through the neighbor-joining method, and the phylogenetic relationships among them were explored. This family was divided into five subgroups, namely, Groups I, II, III, IV, and V, as shown in Figure 3A. The IV Group had 11 members, which was the largest group. Group III was the smallest, with only four BrCC–NBS–LRR proteins. Groups I, II, and V had eight, eight, and six members, respectively (Figure 3A).

The *BrCC–NBS–LRR* gene structures were analyzed on the basis of the alignments between whole cDNA and genomic DNA sequences, which were downloaded from BRAD and blasted in the National Center for Biotechnology Information (NCBI). Then, the structures were drawn with Gene Structure Display Server, as shown in Figure 3B. Nearly half of all *BrCC–NBS–LRR* genes had only one exon (17 of 39), and nine genes had two exons and one intron. However, *Bra026094* had most exons up to 12. The rest of the *CC–NBS–LRR* genes contained more than three exons. Specifically, six members had three exons, two had four, and four had five (Figure 3B). In terms of structure performance, most *BrCC–NBS–LRR* genes have similar exons and introns in the same phylogenetic subfamily.

The conserved protein motifs of BrCC–NBS–LRR were searched on the online tool MEME, which was set to monitor 10 putative protein domains (shown in Figure 3c) with the protein sequences listed in Appendix A and sequence logos of motifs in Appendix A. All the CC–NBS–LRR proteins of Chinese cabbage had the motif 6, which encodes LRR domains. However, the encoding proteins of motif 2, 3, 4, 7, and 10 belong to NB–ARC function domains, which were identified as central nucleotide-binding domains of resistance (R) proteins in plants. Moreover, the motif 3 was found in each BrCC–NBS–LRR protein, and it may be an essential motif for NBSs of Chinese cabbage. Additionally, most BrCC–NBS–LRR proteins contained motif 2 (85%) or motif 4 (82%). All of the 40 BrCC–NBS–LRR proteins contained one or more NB–ARC domain motifs and some other motifs, but Br011432 had the lowest number of motifs. In our searched results, the functions of motifs 1, 5, 8, and 9 were undefined, although motifs 1 and 5 were highly conserved in the CC–NBS–LRR family of Chinese cabbage. Similar motifs were found in the same BrCC–NBS–LRR superfamily, and Groups I, II, and III showed higher degrees of conservation than Groups IV and V.

### 2.5. Cis-Acting Elements Analysis of the CC–NBS–LRR Family Promoter in Brassica rapa

The promoter sequences of *BrCC–NBS–LRR* genes obtained from BRAD were approximately 265–1500 bp upstream of transcriptional start sites (Appendix A). They were scanned with the online tool New PLACE for the prediction of putative cis-acting regulatory DNA elements. The disease-resistant related cis-elements of *BrCC–NB–LRR* promoters were further analyzed and drawn in Figure 4. A total of 12 types of disease-resistant related cis-elements were found, namely, AGCBOXNPGLB, BIHD1OS, ELRECOREPCRP1, GCCCORE, GT1CONSENSUS, GT1GMSCAM4, MYB1LEPR, SEBFCONSSTPR10A, TL1ATSAR, WBOXATNPR1, WBBOXPCWRKY1, and WRKY71OS (Figure 4). They were distributed randomly in their promoters with different items (Appendix A). The cis-elements of GT1CONSENSU were detected in each *BrCC–NBS–LRR* promoter. Especially, in GT1CONSENSU, the GT-1-like factors bind the PR-1a promoter which influences the expression level of SA-inducible genes [23]. GT1CONSENSU had the largest number of cis-elements among the CC–NBS–LRR members of Chinese cabbage (up to 411). *BrCC–NBS–LRR* proteins contained different GT1CONSENSU elements with numbers ranging from 3 to 24. WRKY71OS was the second largest number cis-element of in *BrCC–NBS–LRR* with a total of 206, which are related to the WRKY proteins that bind specifically to TGAC-containing W box elements within the pathogenesis-related class 10 (PR-10) genes [24]. Most *CC–NBS–LRR* members contained WRKY71OS elements, except *Bra019754*. A total of 38 *BrCC–NBS–LRR* members contained GT1GMSCAM4 in their promoters, the expression of which is induced by pathogens and mediated in part by a GT-1 box that interacts with a GT-1-like transcription factor. WBOXATNPR1 and WBBOXPCWRKY1, a salicylic acid (SA)-induced WRKY DNA binding protein and a “W box” WRKY protein that binds specifically to DNA sequence motifs [25], appeared in 31 and 25 *BrCC–NBS–LRR* gene promoters with a total of 96 and 20 items (Figure 4B). A total of 68 BIHD1OS elements were present in 32 *BrCC–NBS–LRR* genes; these elements are the binding sites of OsBIHD1 in disease resistance responses [26]. SEBFCONSSTPR10A, a binding site of the potato silencing element binding factor (SEBF) gene found in the promoter of the pathogenesis-related gene PR-10a, was present in 18 *BrCC–NBS–LRR* gene promoters with a total of 29. Nearly 10 elements were found in other *BrCC–NBS–LRR* gene promoters, such as ELRECOREPCRP1, which contained the consensus sequence of the elements W1 and W2 of PR1-1/PR1-2 promoters, and the WRKY1 protein binding site had 10 items in the 10 promoters of *BrCC–NBS–LRRs*. Five MYB1LEPR cis-elements were found in five *BrCC–NBS–LRR* genes and were related to Pti4(ERF), which regulates defense-related gene expression assisted by GCC box and non-GCC box cis elements [27]. Two GCC-box cores were present in many pathogen-responsive genes on the *Bra026682* promoter, and only one TL1ATSAR was detected in the promoter regions of 13 NPR1-responsive ER-resident genes on the *Bra019754* promoter. Only one AGCBOXNPGLB was conserved in most PR-protein genes on the *Bra037139* promoter. Furthermore, 40 cis-elements were present on *Bra011432* and *Bra034631*, and only five cis-elements were present on *Bra026924* gene promoters listed in Appendix A. These results suggested that the expression of homology *BrCC-NBS-LRR* genes might be regulated by different mechanisms, and these genes play different roles in disease resistance in Chinese cabbage.

### 2.6. Tissue Expression Patterns of CC-NBS-LRR Genes in Chinese Cabbage

To explore the expression patterns of *BrCC–NBS–LRR* genes in the roots, stems, leaves, flowers, and siliques, the total mRNA of the different tissues of the Chinese cabbage cultivar A160 were isolated and reversed-transcribed into cDNAs, which were amplified with specific primers (Appendix A) and normalized with *BrActin*. Expression levels were detected through real-time PCR, and the expression characteristics were analyzed, as shown in Appendix A. The leaves had the highest number of *BrCC–NBS–LRR* genes that exhibited significantly high expression levels. The expression levels of *Bra027097*, *Bra027866*, and *Bra036845* in the leaves were 10-fold those in the other tissues. However, *Bra026094* showed high expression levels in all tissues, especially in the roots, stems, flowers, and siliques. In the siliques, only three *BrCC–NBS–LRR* genes showed prominently enhanced expression levels. These results showed that the *BrCC–NBS–LRR* expression levels in the four tissues varied, indicating that they play different roles in the development of Chinese cabbage.

### 2.7. Expression Profiles of BrCC–NBS–LRR Genes in Response to Phytopathogens

To further analyze the potential functions of *BrCC–NBS–LRR* genes in response to fungal diseases, the inbred A160 lines of Chinese cabbage in the five-leaf stage were inoculated with *Hyaloperonospora parasitica* (DM) or distilled water as a control. They were harvested after 1, 3, 6, 12, and 24 h for short inoculations and 3, 5, 7, 10, and 14 days for long inoculations for the detection of the expression levels of *BrCC–NBS–LRR* genes through real-time PCR. The expression characteristics were analyzed, as shown in Figure 5A,B. After the short inoculations, we found that most (63%) of the *CC–NBS–LRR* family genes were up-regulated after 1 h of inoculation. This result indicated that they quickly respond to DM. For example, *Bra034631* expression was induced and increased 11-fold within 1 h. As for *Bra031482*, the expression was up-regulated 10-fold. The expression level of *Bra026978* increased with inoculation time, showing that *Bra026978* was continuously induced by DM. The expression levels of numerous *BrCC–NBS–LRRs* dramatically changed after 12 h of inoculation. However, the expression levels of *BrCC–NBS–LRRs* were relatively evenly up-regulated 3, 5, 7, and 10 days after inoculation (Figure 5B). In addition, the expression levels of *Bra016782* and *Bra026682* were consistently high at all inoculation periods. However, most *BrCC–NBS–LRR* genes (88%) showed decreased expression 14 days after the inoculation of DM relative to the expression levels detected at day 0 of inoculation (Figure 5B). Prominent diseased plaques were observed in Chinese cabbage leaves after 14 days of inoculation. The expression of Bra019755 was up-regulated 18-fold in 3, 5, 7, and 10 days after inoculation and then declined rapidly to 1.

Expression levels after exposure to *Xanthomonas campestris*, a kind of bacterial BR, were further detected through qPCR. The results are shown in Figure 6A,B. Different *BrCC–NBS–LRR* members were induced at different inoculation times (0, 1, 3, 6, 12, and 24 h (short inoculations) and 3, 5, 7, 10, and 14 days (long inoculations)). Seven of the 40 *BrCC–NBS–LRR* genes (18%) were up-regulated more than twofold after 1 h of inoculation relative to the expression at 0 h, and the expression levels of *Br018245*, *Bra030778*, and *Bra017572* increased 3.5-, 7.3-, and 2.2-fold after 3 h of BR inoculations. Six hours after inoculation, only *Bra016785* showed an increase in expression level. The expression of the others showed no change or was down-regulated. However, more *BrCC–NBS–LRRs* (15 of 40) were up-regulated after 12 h of inoculation. The expression levels of *CC–NBS–LRRs* were smooth and steady compared with those after the short inoculations. *Bra030778* was up-regulated in the long inoculations (3, 5, 7, 10, and 14 days), and the expression levels of *Bra029405*, *Bra019063*, *Bra017572*, and *Bra015597* increased remarkably 3 days after inoculation. The expression levels of *Bra030779* and *Bra015597* remarkably increased 5 days after inoculation. After 7 days of BR inoculation, 17 *CC–NBS–LRR* genes showed up-regulated expression, but 93% of the *BrCC–NBS–LRR* genes showed down-regulated expression 14 days after inoculation. Most of the Chinese cabbages showed numerous disease spots. Compared to long time inoculation, these dramatically differential expressions of *BrCC-NBS-LRR* genes in the 24 h short inoculation suggested that they respond quickly to BR.

### 2.8. Associated Expression of BrCC-NBS-LRR in Different Inbred Lines of Chinese Cabbage in Response to Phytopathogen

To further analyze the expression characteristics of *BrCC-NBS-LRR* in response to fungal and bacterial disease, 24 inbred lines of Chinese cabbage were applied to screen the sensitive and resistant lines to DM and BR. On the basis of statistics (Appendix A), inbred lines A95 and A167 showed the most sensitivity compared to the other 23 lines to DM and BR, respectively, and, in contrast, A24 and A96 were resistant.

Then, the expression levels of inbred lines A95 and A24 were detected when they were exposed to DM for 1 day and 14 days, as shown in Figure 7. The expression levels of most *BrCC–NBS–LRR* genes in the sensitive inbred line A95 and resistant line A24 were up-regulated in response to DM. A total of 11 *BrCC–NBS–LRR* genes showed similar expression characteristics. The expression levels in the two lines were induced by DM to a higher degree than by the control. Moreover, the expression levels in A24 were even remarkably higher than those in A95, not only at 1 day of DM inoculation but also after the control inoculation, particularly those of *Bra011432*, *Bra026368*, *Bra030778*, *Bra026682*, *Bra034631*, *Bra017572*, *Bra027332*, *Bra018245*, *Bra018863*, *Bra019755*, and *Bra037139*. Meanwhile, 13 *CC–NBS–LRR* family members of Chinese cabbage had the same trend of expression as above in the long inoculations (14 days), particularly *Bra013947*, *Bra026368*, *Bra029405*, *Bra030778*, *Bra030779*, *Bra036995*, *Bra027097*, *Bra018863*, *Bra019752*, *Bra019755*, *Bra036845*, *Bra037139*, and *Bra015597*. The trends of *Bra026368*, *Bra030778*, *Bra018863*, *Bra019755*, and *Bra015597* in the 1 day short inoculation were consistent with those in the 14 day long inoculation. By contrast, *Bra009882* and *Bra026978* expression in A24 and A95 decreased obviously after 1 day of DM inoculation.

As for disease stress due to bacterial BR, the sensitive line A167 and resistant line A96 were also inoculated with *Xanthomonas campestris*, and the expression levels of the *BrCC–NBS–LRR* family were detected through qPCR 1 and 14 days after inoculation, as shown in Figure 8. A total of 14 *BrCC–NBS–LRR* genes were induced by BR after 1 day of inoculation in A167 and A96, and six *BrCC–NBS–LRR* genes were induced after 14 days of inoculation. In addition, the expression levels in A96 were even higher than those in A167 after inoculation. Especially, the expression levels of *Bra035424* and *Bra030778* in A167 were up-regulated more than 11-fold. Only one member showed the same expression trend as above (*Bra027866*). However, the expression of *Bra030779* was down-regulated after 1 day of BR inoculation. These results revealed that the *CC–NBS–LRR* gene family of Chinese cabbage may play different roles in response to a disease and the type of role depends on inoculation time and disease types.

## 3. Discussion

Plant disease resistance genes (R genes) have mechanisms for recognizing and providing direct or indirect protection against pathogens in plants and play a critical role in effector triggered immunity (ETI) [28]. Of all five classes of R genes, the NBS–LRR class encodes the largest protein family with one or more NBSs and an LRR domain at the C terminal [29]. The NBS–LRR protein family usually contains two major subfamily *TNL* genes and *non-toll/interleukin-1 receptor-NBS-LRR (nTNL)* genes, which is composed CC–NBS–LRR [30]. In our study, 40 *CC–NBS–LRR* members of Chinese cabbage were searched and identified on the basis of the entire genome of Chinese cabbage. Although 47.1% NBS encodings of *Brassica rapa* appeared to have undergone tandem duplication and to be distributed in tandem arrays [31], all *BrCC–NBS–LRRs* were unevenly distributed on A01–10, with five pairs of two copies (Figure 1) and two pairs of three copies. No *BrCC–NBS–LRR* members were found on A04 and A07; however, A09 had the most members on it with 12 items. *Brassica rapa* and *Arabidopsis thaliana* belong to *Brassicaceae* plants and shared high homology with each other. A total of 21 *AtCC–NBS–LRR* genes were found on the whole genome, 17 members were found on Chr.1, and only one was found on A05 (Appendix A), which showed that the distribution was more extreme. Furthermore, *CC–NBS–LRRs* of Chinese cabbage were expanded 1.9-fold compared to *Arabidopsis thaliana*.

Although *Brassica rapa* shared high homology with model monocotyledonous plant *Arabidopsis thaliana,* based on the protein sequence the phylogenetic tree of the CC–NBS–LRR family was built to explore the homology among *Arabidopsis thaliana*, *Brassica rapa* and monocotyledon rice. To our surprise, not all these BrCC–NBS–LRRs were highly clustered in the same subgroup with their At orthologs; moreover, CC–NBS–LRRs of the same species were clustering together. Even two BrCC–NBS–LRRs shared high homology with rice instead of *Arabidopsis*, such as Bra013213 and Bra017572, as shown in Figure 1.

The protein structures of BrCC–NBS–LRRs were usually composed of three parts: one or more NBSs, an LRR domain at the C terminal, and a CC domain at the N terminal. NBS-encoding domains are necessary for the recognition of diverse pathogens in plants [32]. Five kinds of NBS-encoding motifs were found in BrCC–NBS–LRRs, such as motifs 2, 3, 4, 7, and 10 (Figure 3), and motif 3 was the most conserved in the BrCC–NBS–LRRs. LRR may be a core function domain to the recruitment of protein [33]. LRRs in general are irregular and variably responsible for interactions between resistance proteins on account of a hydrophobic backbone for three-dimensional integrity and many free solvent-exposed residues of the LRR domain [34,35]. The CC domains were detected in all the CC–NBS–LRR members of *Brassica rapa* on IntrePro. The CC motif was analogous to the LRR motif in the critical core of the three-dimensional structure composed of hydrophobic chains and free solvent-exposed residues. The CC domain of a *CC–NBS–LRR* gene of LR10 in wheat directly recognizes pathogens; this result indicated that the CC domain function of *CC–NBS–LRRs* needs to be further explored and excavated [36].

The cis-acting elements of gene promoters were essential for transcriptional expression, and the types of cis-acting elements indicated the potential functions of genes, particularly in response to pathogens [37]. All the *BrCC–NBS–LRR* promoters were downloaded from BRAD and analyzed with an online tool New PLACE for the searching of cis-acting regulatory elements. A total of 12 disease-resistance-related cis-acting elements were found in the promoters of BrCC–NBS–LRR. The GT1CONSENSU (consensus GT-1) elements dominated as investors and appeared on the promoters in tandem. It was first reported as the binding sites of many light-regulated genes in many species, for example, *ribulose bisphosphate carboxylase small chain*, *phytase* gene [38,39]. The binding of GT-1-like factors to the *PR-1a* promoter can reduce TMV infection and SA treatment and influence the expression level of SA-inducible genes [23]. The function of GT1GMSCAM4 was analogous to that of GT1CONSENSU. WRKY71OS, WBOXATNPR1, and WBBOXPCWRKY1 are typical disease-related elements found in BrCC–NBS–LRRs. WRKY71OS is not only involved in plant defense signaling by binding specifically to W box elements of the PR-10, but also plays a role in GA and ABA signaling pathways [25,40]. “W-box” sequences were found in the promoters of WBOXATNPR1 and WBBOXPCWRKY1 to bind with NPR1 and PR1, participating in plant defense response [41,42]. A total of 68 BIHD1OS elements were found in *BrCC–NBS–LRR* genes, that is, the binding site of the BELL-type transcription factor of OsBIHD1 associated with resistance and response in rice [26]. Some minor cis-acting elements are associated with disease resistance, such as SEBFCONSSTPR10A, ELRECOREPCRP1, MYB1LEPR, TL1ATSAR, GCCCORE, and AGCBOXNPGLB.

In recent years, a large number of plant genomes have been sequenced, and an increasing number of NBS-encoding families of different species have been analyzed and surveyed [36]. In addition, five *Brassicaceae* NBS–LRR genes were comprehensively analyzed and compared on the basis of established cross-species phylogenetic and syntenic relationships of NBS genes for the study of the R gene in *Brassicaceae*, particularly in *Arabidopsis lyrata*, *Arabidopsis thaliana*, *Brassica rapa*, *Capsella rubella* (127), and *Thellungiella salsuginea* [43]. *NBS-LRR* genes exert critical effects in response to multiple pathogens, including viruses, bacteria, and fungi [44]. CC–NBS–LRR, as a main subfamily of NBS–LRR, is involved in disease resistance in plants, such as *Arabidopsis thaliana*, rice, and wheat. However, they are rarely reported in Chinese cabbage in terms of response to DM and BR. In our study, the A160 inbred lines of Chinese cabbage were inoculated with *Hyaloperonospora parasitica* and *Xanthomonas campestris*, and short and long inoculations were performed for exploring the expression profile. In the short inoculations, heatmaps were drawn after 1, 3, 6, 12, and 24 h of inoculation (Figure 5 and Figure 6). We found that most *BrCC-NBS-LRR* genes were induced drastically 1 h after inoculation, and a peak in expression level occurred 12 h after inoculation with *Hyaloperonospora parasitica* and *Xanthomonas campestris*. *Bra026978* was continuously induced by *Hyaloperonospora parasitica*. The expression levels of *Bra018834*, *Bra019755*, and *Bra026094* were up-regulated by *Hyaloperonospora parasitica* at each inoculation time in the short inoculations, which showed that these genes have a relationship with DM defense. *Bra016785* and *Bra027332* had high expression levels during BR inoculation. For long inoculations, the expressed *BrCC-NBSL-RRs* after inoculation with DM were more abundant than those observed after BR inoculation. Moreover, the differentially expressed genes responding to each pathogen slightly varied. *Bra030778* was induced markedly at each inoculation of BR, but no change in response to DM was observed, revealing that *Bra030778* preferred involvement in BR defense.

An increasing number of CC–NBS–LRRs have been reported, such as Lr10, which is involved in (CC–NBS–LRR)-mediated resistance [45]; Yr10 (CC–NBS–LRR), which plays a key role in resistance to *Pst* [46]; *Lr34* (CC–NBS–LRR), which enhances resistance during adult stages; TaRPM1, which is involved in resistance to *Pst* [47]; and Pm21 from the wild species, which shows high resistance to Bgt [48]. Additionally, in rice Pb1 shows resistance to rice blast only during adult stages [49]. To further identify the function of *BrCC-NBS-LRRs*, the expression characteristics were carried out on the sensitive inbred lines A95 and A167 and resistant lines A24 and A96 to DM and BR for 1 day and 14 day inoculation, as shown in Figure 7 and Figure 8. There were 11 *BrCC-NBS-LRR* genes showing similar expression profiles induced by DM rather than the control; moreover, A24 was even higher than A95 DM for 1 day inoculation and for 13 for the long inoculation (14 days). The expression levels of *BrCC–NBS–LRR* genes after 1 day of inoculation with DM and BR were compared. *Bra030778* and *Bra034631* were induced and had the same expression level. In the long inoculation of DM and BR, three *BrCC–NBS–LRRs* (*Bra013947*, *Bra019755*, and *Bra015597*) showed higher expression levels. These results revealed that *BrCC–NBS–LRR* family genes play various roles in response to phytopathogens, and the roles depend on disease type and inoculation time.

## 4. Materials and Methods

### 4.1. Plant Materials and Inoculation

A total of 24 inbred lines of Chinese cabbage from Northeast Agricultural University were cultivated in a greenhouse subjected to day (28 °C)/night (18 °C) and 16 h light/8 h dark cycles. For the expression profiles of DM and BR, the *Brassica rapa* inbred line A160 of the fifth leaf was sprayed with pathogens and harvested 3, 5, 7, 10, and 14 days after infection, and non-infected Chinese cabbage was used as the control. Additionally, for screening the sensitive and insensitive plants, the fourth leaves of 24 inbred lines were infected by both *Hyaloperonospora parasitica* and *Xanthomonas campestris* for 14 days, respectively, until invasion of the two diseases with water as the control. The whole leaf and lesion areas of Chinese cabbage were surveyed, and three replications of biological duplication were carried out. The lesion areas of leaves were measured, and according to statistics, the inbred lines with a lesion area taking up greater than 75% of the whole leaves were sensitive lines, and at the same time less than 20% was used as the resistant line. Inbred lines A95 and A167 were the most sensitive to DM and BR, respectively, whereas A24 and A96 were resistant. The leaves of the sensitive and resistant plants were harvested and frozen in liquid nitrogen and then stored at −80 °C for RNA isolation.

The inbred line A160 of Chinese cabbage was grown in a greenhouse until four leaves were moved to light incubator with 6°C for 25 days of vernalization, and then they were grown under conventional environment until flowering. In total, the tissues of root, stem, leaf, flower and silique were harvested, respectively, for RNA isolation.

### 4.2. Identification and Analysis of CC–NBS–LRR in Chinese Cabbage

All the CC–NBS–LRR members of Chinese cabbage were obtained from BRAD (http://brassicadb.org/brad/, accessed on 27 February 2021), with the *B. Rapa* genome (version 1.5). They were identified with the NB–ARC domain according to the Hidden Markov Model of pfam00931 and the LRR domain of PLN03210 at the C terminal with the BLAST program of the NCBI database (https://blast.ncbi.nlm.nih.gov/Blast.cgi, accessed on 14 April 2021). The CC domain of BrCC–NBS–LRRs was confirmed using MPI Bioinformatics Toolkit (https://toolkit.tuebingen.mpg.de/#/tools/quick2d, accessed on 27 February 2021. 2008–2021) [50]. The other CC–NBS–LRR sequence of *Arabidopsis thaliana*, with *A thaliana* genome: Araport11 and *Oryza sativa* were downloaded from TAIR (http://www.arabidopsis.org/, accessed on 27 February 2021. 2005) and Rice Genome Annotation Project (http://rice.plantbiology.msu.edu/, 6 February 2013), which were confirmed using the Basic Local Alignment Search Tool of NCBI. InterPro (http://www.ebi.ac.uk/interpro/scan.html, accessed on 27 February 2021. 2021) and Pfam (http://pfam.xfam.org/search#tabview=tab1, accessed on 27 February 2021. 2018) were used for further confirmation [51].

### 4.3. The Cis-Acting Elements in Promoters, Chromosomal Locations, and Gene Structures

The promoter sequences of *BrCC–NBS–LRR* genes were searched on BRAD (http://brassicadb.org/brad/, accessed on 27 February 2021), and all of them were analyzed using New PLACE (https://sogo.dna.affrc.go.jp/cgi-bin/sogo.cgi?lang=en&pj=640&action=page&page=newplace, accessed on 27 February 2021. 2011–2021), which is a database containing plants used in the analysis of cis-acting regulatory elements [52]. Information on chromosomal location was obtained from BRAD and assessed by MapInspect. The genome and CDS sequences of CC–NBS–LRR were downloaded from BRAD, which were placed in the Gene Structure Display Server (http://gsds.cbi.pku.edu.cn/, accessed on 27 February 2021. 2021) for the reconstruction of the gene structure [53].

### 4.4. Phylogenetics and Conserved Motif Analysis of CC–NBS–LRR Protein

The CC–NBS–LRR protein sequences of Brassica rapa and *Arabidopsis thaliana* and *Oryza sativa* were downloaded from BRAD, TAIR, and Rice Genome Annotation Project. First, multiple sequences were aligned with MEGA6, then the phylogenetic trees were calculated and constructed using the neighbor-joining method within 1000 replicates [54].

The protein sequences were analyzed using the online tool MEME Suite with the following settings: site distribution, any number of repetitions; number of motifs, 10.

### 4.5. Total RNA Extraction and Real-Time PCR Analysis

The plants described in the Plant Materials section were harvested for RNA extraction with TRIzol™ reagent (Invitroge, Thermo Fisher, Waltham, MA, USA), and 2 mg of RNA was reversed-transcribed to cDNA with TransScript One-Step gDNA Removal and cDNA Synthesis SuperMix kits (TransGen, Beijing, China). Approximately 15 L of total volume was used for real-time PCR, containing 1 L of specific primers, 3 L of cDNA samples diluted 30–50-fold and 7.5 L of SYBR Green real-time PCR master mix (TOYOBO, Osaka, Japan). The reactions were amplified using a qTower3G (Analytik Jena, Jena, Germany) real-time PCR detection system for 40 cycles with unimodal dissolution curve and tubulin as an internal reference [55]. The primers are listed in Appendix A.

## Figures and Tables

**Figure 1 ijms-22-04266-f001:**
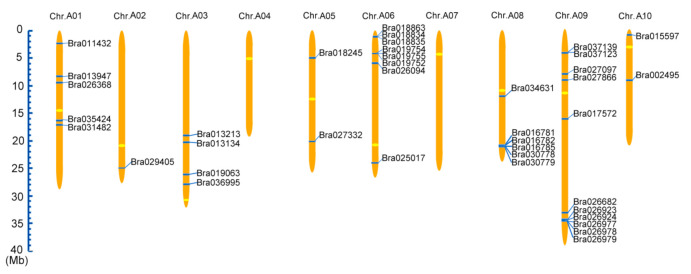
The chromosomal location of *CC–NB–LRR* genes from Chinese cabbage. The scale represents 40 Mb chromosomal distance. The chromosome numbers are labeled on the top of them. The duplicate genes are connected with red line.

**Figure 2 ijms-22-04266-f002:**
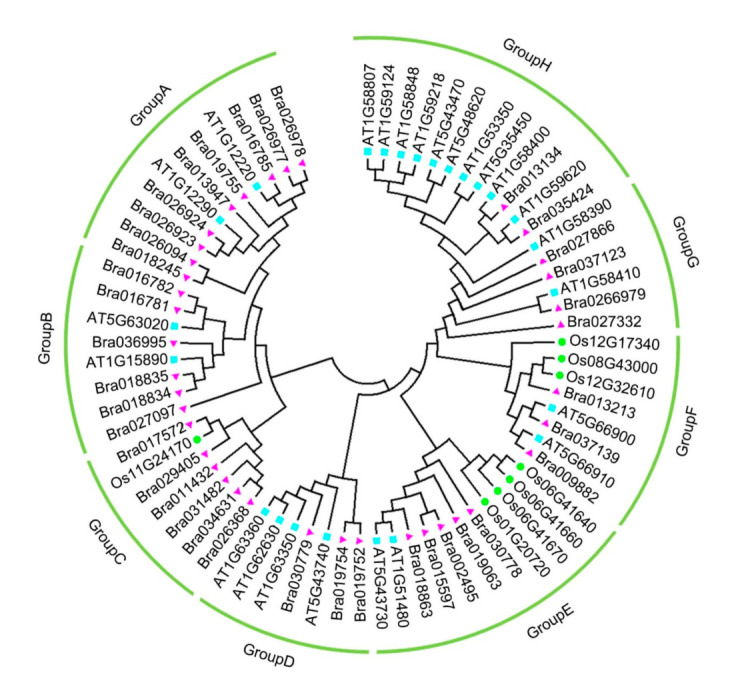
Phylogenetic relationships of CC–NB–LRR protein among *Brassica rapa*, *Arabidopsis thaliana* and *Oryza. sativa*. The unroot phylogenetic tree was constructed by MEGA 7.0 by neighbor-joining method with 1000 bootstrap replicates. CC–NB–LRRs of different plants are indicated with different colors and shapes. Pink triangles represent *Brassica rapa*, green circles represent *Oryza. sativa* and blue squares represent *Arabidopsis thaliana.*

**Figure 3 ijms-22-04266-f003:**
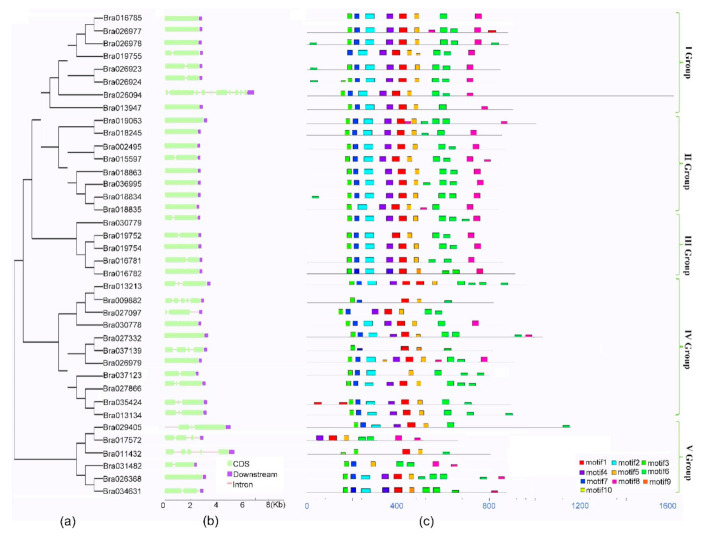
The Phylogenetic tree, gene structure and multiple motifs of CC–NBS–LRR family in Chinese cabbage. (**a**) Phylogenetic tree was constructed based on the protein sequence of Chinese cabbage by MAGE7.0; (**b**) The gene structure of Chinese cabbage *CC–NBS–LRRs*. Exon and intron are represented by green boxes and lines; (**c**) The multiple conserved motifs of Chinese cabbage CC–NBS–LRR proteins. Different colored boxes represent function motifs, identified by an online tool MEME. Motifs 2, 3, 4, 7, 10 were NB-ARC function domains.

**Figure 4 ijms-22-04266-f004:**
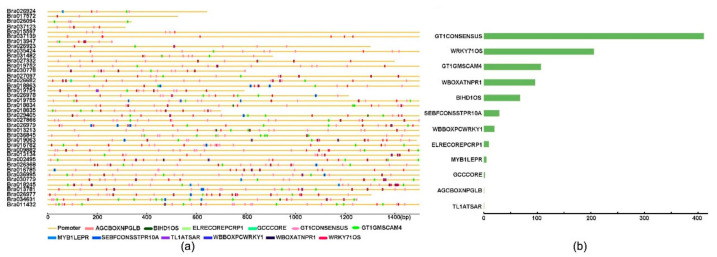
The disease resistant related cis-elements on *CC–NB–LRR* promoters of Chinese cabbage. (**a**) The distribution of disease resistant related cis-elements on *BrCC–NB–LRR* promoters; (**b**) The total number of each disease resistant related cis-element on Chinese cabbage *CC–NB–LRR* promoters.

**Figure 5 ijms-22-04266-f005:**
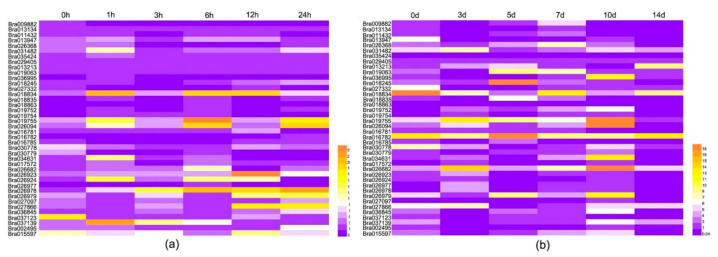
Expression patterns of *BrCC–NBS–LRRs* in response to DM. (**a**) The expression levels of *BrCC–NBS–LRR* genes in response to DM with short inoculation time for 0, 1, 3, 6, 12 and 24 h; (**b**) the expression levels of *BrCC–NBS–LRR* genes in response to DM with long inoculation time for 0, 3, 5, 7, 10 and 14 days.

**Figure 6 ijms-22-04266-f006:**
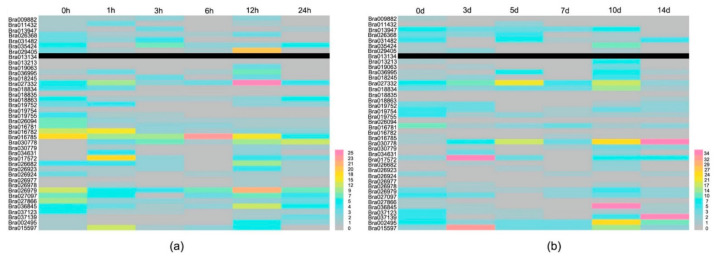
Expression patterns of *BrCC–NBS–LRRs* in response to BR. (**a**) The expression levels of *BrCC–NBS–LRR* genes in response to BR with short inoculation time for 0, 1, 3, 6, 12 and 24 h; (**b**) the expression levels of *BrCC–NBS–LRR* genes in response to BR with long inoculation time for 0, 3, 5, 7, 10 and 14 days.

**Figure 7 ijms-22-04266-f007:**
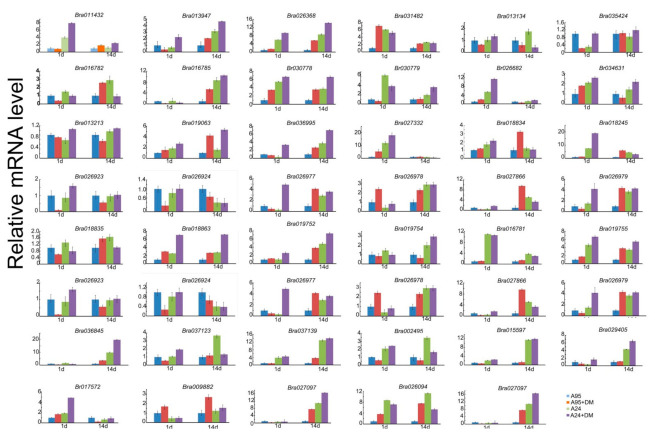
The *BrCC–NBS–LRR* expression levels of inbred line A95 and A24 with DM inoculation. The *CC–NBS–LRR* genes’ expression levels were detected in A95 and A24 for both 24 h (short inoculation) and 14 days (long inoculation) of DM inoculation with water as control. They represent results performed in triplicate. Error bars represent ± SE.

**Figure 8 ijms-22-04266-f008:**
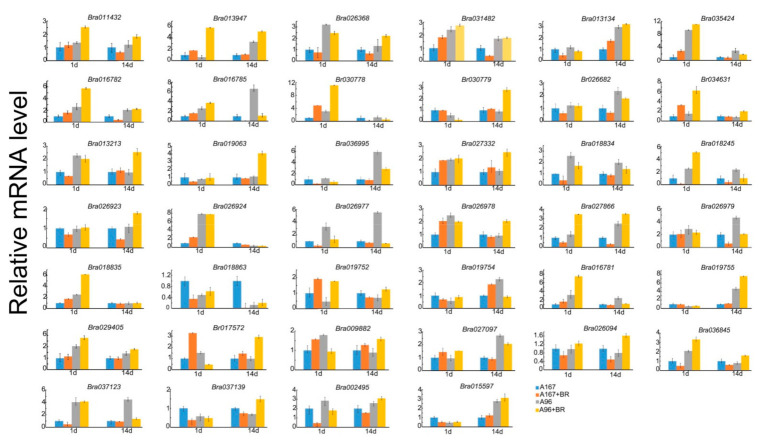
The *BrCC–NBS–LRR* expression levels of inbred line A95 and A24 with BR inoculation. The *CC–NBS–LRR* genes’ expression levels were detected in A95 and A24 for both 24 h (short inoculation) and 14 days (long inoculation) of BR inoculation with water as control. They represent results performed in triplicate. Error bars represent ± SE.

**Table 1 ijms-22-04266-t001:** *CC–NB–LRR* genes in Chinese cabbage.

Gene ID	Genome Position	CDS Length(bp)	Exon	Protein Length(aa)	Molecular Weight (kDa)	Isoelectric Point
*Bra011432*	A01:2,191,374...2,196,276	2424	5	807	91.25906	5.96
*Bra013947*	A01:8,643,759...8,646,482	2724	1	907	103.78083	5.79
*Bra026368*	A01:9,720,079...9,723,012	2934	1	977	110.51354	6.56
*Bra031482*	A01:17,037,929...17,040,208	2145	2	714	81.85303	8.46
*Bra035424*	A01:16,404,803...16,407,820	2703	3	900	104.13992	6.56
*Bra029405*	A02:25352667...25357391	3489	4	1162	135.16142	6.22
*Bra013134*	A03:20,277,197...20,280,164	2736	3	911	104.87017	8.30
*Bra013213*	A03:19,840,894...19,844,167	2916	3	971	111.04147	7.81
*Bra019063*	A03:26,487,398...26,490,427	3030	1	1009	114.15502	6.30
*Bra036995*	A03:29,029,474...29,032,056	2583	1	860	97.34071	6.87
*Bra018245*	A05:6,909,119...6,911,701	2583	1	860	98.19411	6.50
*Bra027332*	A05:20,364,432...20,367,560	3129	1	1042	119.33079	8.42
*Bra009882*	A06:17,846,911...17,849,759	2466	5	821	93.52592	6.33
*Bra018834*	A06:1,698,860...1,701,415	2556	1	851	96.87193	5.85
*Bra018835*	A06:1,694,120...1,696,651	2532	1	843	95.76674	6.08
*Bra018863*	A06:1,554,378...1,556,936	2559	1	852	96.60849	5.99
*Bra019752*	A06:4,608,614...4,611,267	2583	2	860	98.23598	8.12
*Bra019754*	A06:4,599,425...4,602,097	2673	1	890	100.98419	6.63
*Bra019755*	A06:4,595,072...4,597,754	2586	2	861	98.09356	6.05
*Bra026094*	A06:6,170,240...6,176,552	4842	12	1613	182.73632	6.22
*Bra016781*	A08:19,944,556...19,947,218	2589	2	862	98.25993	5.64
*Bra016782*	A08:19,949,435...19,952,182	2748	1	915	104.28085	5.44
*Bra016785*	A08:19,969,925...19,972,603	2679	1	892	101.87473	6.10
*Bra030778*	A08:20,402,328...20,404,925	2598	1	865	98.73556	5.14
*Bra030779*	A08:20,398,567...20,401,178	2517	2	838	95.04727	5.82
*Bra034631*	A08:12,381,249...12,384,097	2655	3	884	99.63007	6.46
*Bra017572*	A09:16,599,373...16,602,112	2001	5	666	76.24774	5.48
*Bra026682*	A09:33,197,405...33,200,004	2514	2	837	95.01493	6.02
*Bra026923*	A09:34,338,460...34,341,119	2559	2	852	97.16891	6.09
*Bra026924*	A09:34,342,578...34,345,222	2544	2	847	96.53494	5.73
*Bra026977*	A09:34,580,053...34,582,713	2661	1	886	101.12124	5.72
*Bra026978*	A09:34,584,346...34,587,018	2673	1	890	101.55526	6.14
*Bra026979*	A09:34,589,539...34,592,283	2745	1	914	103.82301	6.74
*Bra027097*	A09:8,445,238...8,447,974	1992	3	663	75.57687	6.33
*Bra027866*	A09:9,368,958...9,371,944	2790	3	929	106.98517	6.51
*Bra036845*	A09:25,444,116...25,448,953	3876	5	1291	146.91249	6.22
*Bra037123*	A09:4,366,953...4,369,453	2415	2	804	93.69916	8.98
*Bra037139*	A09:4,297,244...4,300,267	2454	4	817	93.03288	5.42
*Bra002495*	A10:9,233,853...9,236,492	2640	1	879	99.47706	8.46
*Bra015597*	A10:747,361...750,005	2571	2	856	97.89404	7.43

## Data Availability

This statement if the study did not report any data.

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
