# Peer review of "Genome-Wide Identification and Analysis of CC-NBS-LRR Family in Response to Downy Mildew and Black Rot in Chinese Cabbage"

_ijms, 2021, doi:10.3390/ijms22084266_

Round 1
Reviewer 1 Report
Please revise the draft. There are many errors seen elsewhere and makes difficult to read.

Author Response
Dear Reviewer,
Thank you for all the Comments and Suggestions for our manuscript entitled “Genome-Wide Identifification and Analysis of CC-NBS-LRR family in response to Downy Mildew and Black Rot in Chinese Cabbage” , that was a great help for our study and further research. We revised point by point in MS

Reviewer 2 Report
The paper is well written and scientifically sound, but some improvements can be made to my opinion:
Please indicate genome versions (B rapa & A thaliana) in the result part.
Figs5 & 6 are not readable.
Fig7: the figure is quite difficult to read. Can it be displayed in a more convenient way?
Please provide additional information for Arabidopsis regarding the expression levels of Br orthologs, e.g. using the BAR website.
qPCR: please provide evidences that primers are really specific of each target, because NBS genes may have large similar parts.
Author Response

(The authors gave the same response as above.)

Reviewer 3 Report
The authors performed a good job in the experimental part, with a manuscript with potential to be published. However, the information should be summarized and reorganized in order to enhance the quality of the manuscript. In addition, there are many concerns about the manuscript that request minor revisions, as it is described below:
- In the tittle authors should be correct the word “identifification” (line 2)
- An abbreviation list must be added in order to clarify the large amount of abbreviations without explanation along the manuscript, such as NB-ARC domain (line 34), PI (line 78), SA (line 216), ETI (line 364), etc.
- The sentence “A NBS-LRR type R gene of Pi-ta played a derict role in Magnaporthe grisea of rice by interaction of the effector AVR-Pita[12]. And RRS1 protein of Arabidopsis thaliana interacted directly with PopP2, which was the pathogen protein of bacterial wilt[13]. RPS2 and RPM1 resistance genes from Arabidopsis were in response to Pseudomonas syringae by indirect interaction with AvrRpm1 and AvrB[14,15].The ectopic overexpression of Arabidopsis RPW8 enhanced the resistant ability to powdery mildew in grapevine[16]” (lines 53- 59) give complex information that is not relevant for the study, so it should be removed.
- Last part of the introduction section should be summarized, and most relevant information should be highlighted to clarify the aims and the novelty of this study.
- The information of the sentence “Arabidopsis thaliana is a dicotyledonous model plant, whereas Oryza sativa is a typical monocotyledonous plant” (lines 135-136) should also be explained in the introduction.
- In table 1 the molecular weight should be expressed in kDa instead of Da.
- There are many references to scientific names that should be written in italics (lines 152, 156, 350, 436, etc)
- The quality of the images must be improved in some figures because they are almost impossible to read. For example, figure S1, figure 4, figures 5 and 6, and figures 7 and 8 (related to the mRNA expression levels). The size of the graphs and the words must be expanded to allow reading the results.
- In figure 4B and figure S3 the units should be added to the X axis.
- In line 270 the disease caused by the Hyaloperonospora parasitica should be added to clarify the aim of this part.
- In line 402, the sentence “It first reported as the binding…” should be changed by “It was first reported that as the binding…”
- In line 375, the name of Arabidopsis thaliana should be corrected.
- The sentence “The NBS–LRR protein family usually contains two major subfamily TNL genes and non-toll/interleukin-1 receptor-NBS-LRR (nTNL) genes, which is composed CC–NBS–LRR”. (lines 365-367) is confused, so it should be changed in order to specify that CC-NBS-LRR genes are presented and studied in Chinese cabbage pathogen response.
- A conclusion part should be added in order to summarize the results and facilitate the information obtained after the analysis.
Author Response
Dear Reviewer,
Thank you for all the Comments and Suggestions for our manuscript entitled “Genome-Wide Identifification and Analysis of CC-NBS-LRR family in response to Downy Mildew and Black Rot in Chinese Cabbage” , that was a great help for our study and further research. We revised point by point.

Reviewer 4 Report
The manuscript entitled Genome-Wide Identification and Analysis of CC-NBS-LRR family in response to Downy Mildew and Black Rot in Chinese Cabbage aims to present a new insight on the Chinese cabbage CC-NBS-LRR genes. These are considered as one of the main sub-families of NBS-LRR genes involved in disease resistance in plants, and the authors successfully analysed their response to Downy Mildew and Black Rot. This information, as pointed by the authors, is important for further use on the research on the role of these genes in Chinese cabbage.
Although the work developed is interesting and valuable, the manuscript reveals major concerns regarding the way the study is presented.
The introduction is well written, the context of the research is presented clearly, and the citations seem adequate. Nonetheless, the remaining points of results, material and methods and specially the discussion need work. Also, the authors should take into consideration that throughout the whole document the space before the citation brackets are missing.
In the results section, it is evident the mix of material and methods and results, for most of the sub-sections. This is evident by all the citations used when introducing the results for each sub-section. Citations are to be avoided on the results and should be presented on material and methods.
Material and methods section is lacking a lot of information and should be improved, but I will develop this point by point.
The discussion appears to be the weakest point on this manuscript. The discussion presented, in my opinion, simply does not properly integrate the full scale of the results and does not really discuss them. Most of this section seems more of a final resume of the results, as is evident from line 425 until the end, where most of the text is just a second description of the results. I strongly believe that a major rewriting of this section is deeply needed.
Species names are not all in italics throughout the manuscript. Please revise.
Please find below the comments and concerns, point by point:
P1 Line 13: space missing in effector triggered
P2 Line 50: The (9) after monocotyledons is the number of genera or is it a reference?
P2 Line 85: pv. Is missing for Xanthomonas campestris
P3 Line 96: The authors say a total of 41 CC-NBS-LRR family candidates where found, but further on state they only screened 40. Why was one candidate dropped?
P4 Line 113: Why do the authors use the nomenclature of A01, A02 … A04 for the chromosomes while on the figure use Chr. 1 Chr.2, etc. Please use the same for all the text.
P4 Line 121: I think the sentence should read: … A3, A01, A08 respectively.
P4 Line 127: Chinese is misspelled
P5 Line 139: The authors say that MEGA7.0 was used when on material and methods and the caption for figure 2 state that MEGA6.0 was used. Please correct the version of the program.
P5 Line 156: Arabidopsis thaliana should be in italics.
P6 Line 163: Please insert which colour and form corresponds to each species
P7 Line 183: This paragraph appears a bit confusing. The authors should try to simplify the text in order to facilitate reading. They could include not only the presence of the motifs but also in which proteins are absent. For example, Bra011432 could also be described by the low number of motifs.
P7 Line 184: Figure 3C and not Figure 3.
P9 Line 253: This whole sub-section (2.6) appears on the document without being previously referred anywhere else. This part of the work is not referred neither on the abstract, nor on the introduction or the material and methods. If the authors want to present these results they should be introduced previously on the manuscript. Also, please clarify the use of inbred line A160?
P9 Line 257: Were it is Table S4 it should be Table S6. Also, this table is lacking some information. I believe that information as anneling temperature, amplicon size and primer efficiency should be presented on the referred table.
P9 Line 272: The use of the term treatments may not be advisable. What the authors consider as treatments, should considered as inoculations or plant challenged with the pathogen.
P9 Line 273: Heatmaps and not hot maps. This term should be revised throughout the whole manuscript.
P9 Line 282: the number Bra016782 is repeated
P9 Line 285: No previous description is given on these disease plaques. What are they? Please insert a description
P10 Line 320: The authors state that on the basis of statistics inbred lines were selected as sensitive and resistant. What statistics are these? The authors say on material and methods that lines showing lesion are above 75% were considered susceptible while below 20% were considered resistant. How was this are measured, and how was this calculated. This information should be shown.
P10 Line 327: similar expression profile instead of characteristics.
P12 Line 361: As referred before, I believe this whole section needs a big revision and rewriting. The discussion does not clearly integrate all the results and is not clear as how this work and results correlate with the few citations found along this part of the document. This is even more evident on the last part of the discussion regarding gene expression, where most of the text is simply restating the results presented before.
P12 Lines 373-374: The authors here are, again, not consistent on the nomenclature used for the chromosomes. They start with Chr. A4 and finish with A09. Please correct throughout the whole manuscript. Also, this information is not according to figure 1, where chromosome 1 is referred as Chr. 1.
P14 Line 466: The title should not only be plant materials but should also include inoculation.
P14 Line 470: The leaf was sprayed with pathogens. How were these pathogens obtained? How was the inoculum grown and prepared for the inoculation? How were the inoculations preformed and what concentrations were used? The control plants were mock inoculated with water? All this information is important to be presented on the manuscript.
P14 Line 472 – 473: the authors say that the 4th leaves were infected by both Hyaloperonospora parasitica and Xanthomonas campestris. Both pathogens were inoculated on the same leaves? Or inoculations were preformed separately? This needs to be clear on material and methods.
P14 Line 476: As referred before, the authors say, “according to statistics”. What statistics were used? How did the lesion area of the leaves was calculated? This should be clarified.
P15 Line 512: On the section 4.5 several issues should be addressed. The authors preformed a gDNA removal (as part of the RNA extraction kit) but was the presence of gDNA tested in crude RNA? Concerning the qPCR the authors do not indicate the number of cycles, nor how the efficiency of the primers was determined (standard curve, LineReg software?). Were melting curves used to determine the PCR reaction specificity? Also, what formula was used to determine the expression profile (Livak, Hellmans, or other)? A reference should be included to support the choice of the internal reference (tubulin gene).
P15 Line 521: It is not Table S1 but Table S6
Author Response

(The authors gave the same response as above.)

Round 2
Reviewer 4 Report
The authors of the paper entitled “Genome-Wide Identifification and Analysis of CC-NBS-LRR family in response to Downy Mildew and Black Rot in Chinese Cabbage” addressed most of the suggestions made during the first round of review. Nevertheless, I believe that the importance of a detailed description of all the steps of the work preformed should not be overlooked. Therefore, I will suggest that additional information should be added to the material and methods sections.
The authors addressed the all the comments made on the material and methods section but did not incorporate them into the manuscript. So, I leave here my suggestions point by point:
On table S6 the authors added the note “The anneling temperatures ranged from 52℃ to 56℃; the amplicon sizes ranged from 85 to 105bp”. I keep the suggesting of adding two columns to the table describing this information primer by primer.
Regarding the information on the statistics used for the selection of inbred lines, I appreciate the answer of the authors. I suggest that the table with the statistics to be included as supplementary data.
On page 17, line 517, point 4.1 of material and methods, I suggest the inclusion of the information provided by the authors on the cover letter addressing the review suggestions, more precisely, on points 19, 24 and 25.
On point 4.5 of material and methods, I suggest more information such as, if the presence of gDNA was tested in crude RNA, how the efficiency of primers was determined, and what formula was used to determine the expression profile (Livak Hellmans, or other).
Finally, I suggest that a small conclusion should added to the end of the discussion, in order to close and summarize the results of the good work presented.
Author Response
Dear Reviewer,
Our sincere thanks you for suggestions of our manuscript entitled “Genome-Wide Identifification and Analysis of CC-NBS-LRR family in response to Downy Mildew and Black Rot in Chinese Cabbage” in the second round. Now we will show our interpretation point by point.
- On table S6 the authors added the note “The anneling temperatures ranged from 52℃ to 56℃; the amplicon sizes ranged from 85 to 105bp”. I keep the suggesting of adding two columns to the table describing this information primer by primer.
Response:The table S6 were revised as below:
|
Gene No. |
Forward Sprimer Sequence (5’-3’) |
Reverse primer Sequence (5’-3’) |
Ta |
Product |
|
Bra011432 |
TCTTTTTTCTTTTCGGTCGA |
CGCAGTGAGAGTGACATGGT |
53.3 |
81 |
|
Bra013947 |
AGGGAGGTTTATGCGTCGG |
AGTTTGTACTCAGCACAACCTAAGC |
57.6 |
81 |
|
Bra026368 |
TAAGTCCTTGAAATCCAGGCTAAG |
TCTAAACCTTCCTTTTCGAGCA |
55.1 |
81 |
|
Bra031482 |
CAACAAGGTTTGGAACAATTTTT |
AAGTTGCATACCCCTGATCGT |
53.1 |
86 |
|
Bra035424 |
TTTCCAGGAGCAGGTGATTC |
GGATTCAATCCACAAATCGTTG |
54.7 |
91 |
|
Bra029405 |
TGGGATTCAGGTGCAAGAG |
AACGGTAAAAGTGCCACTGC |
55.3 |
81 |
|
Bra013134 |
TATGAGTACACTAATTCTGTAAATT |
CACTAGAGGAAATCCAGGACG |
53.6 |
84 |
|
Bra013213 |
AGTGCTTTGATATATTGCATTTTGC |
TTTACAGAATTAGTGTACTC |
50.1 |
88 |
|
Bra019063 |
TGTGAAGACCATATAAAGTCAGCA |
TGAGGCACCAACGACTTCTTA |
55 |
90 |
|
Bra036995 |
AGAATGATCAAAGTCTTCATATGGAA |
CTTTGCTCCAATACGACGAA |
53.3 |
87 |
|
Bra018245 |
CAAGGATCTCAGGGACGATT |
CTCTGAACAGTTGCGAGCC |
56.3 |
85 |
|
Bra027332 |
TAGAGAGGCTCAACACTGCG |
CCGCTGGGTTTGTAATCTGT |
56.4 |
81 |
|
Bra009882 |
AAGTAATTGCGCGGCAAGTG |
GCTTCCGCTCAATGGCTAAG |
56.4 |
86 |
|
Bra018834 |
TGGCTTACTATTTTCTCACCAAGTC |
ACTGCTTTGGTGAATGAACAAG |
55.1 |
88 |
|
Bra018835 |
CGGTCTTACTAGCTGGAGGC |
GAATGGACAAGAGAAAGAGGACA |
57.7 |
95 |
|
Bra018863 |
CCGGATATTCCCACACTTG C |
GGCTTTGCCAATCACATTGA |
55.4 |
98 |
|
Bra019752 |
TGTTAAGTCATGTTTCCTCTATTGC |
CCTCACCGATCCAATATTTAATTA |
53.7 |
81 |
|
Bra019754 |
CGCCACGCAGTTAAGACTTT |
GCTATACTTCAAAACTGGAAGAA |
53.9 |
81 |
|
Bra019755 |
CATTGCGAAAAAACTAGGCCT |
ACGTTGTGGATGTCAAGAGCT |
54.6 |
82 |
|
Bra025017 |
TTGCATCAAACCAAACACTGA |
CCCATCAACTGTTGCTTTAGC |
53.6 |
99 |
|
Bra026094 |
TGGCTGCTTATTCGGTGATG |
AGGTTTGCTTCCATTGTGTGAA |
54.7 |
84 |
|
Bra016781 |
CAGATAATAGAGGCATACAGCCATT |
TTTTAAGCTCTAATAAGCATGC |
54.1 |
81 |
|
Bra016782 |
TACTGACCGAAGAGGTAGAAGGA |
CCACATCATGCATTTTCACC |
55.6 |
86 |
|
Bra016785 |
AGGTTGATTTTAAGTACCTTGAGGA |
TGCTGAGCCTACGGAGATTG |
56.1 |
81 |
|
Bra030778 |
AGGTGGTGGCTAGGAGGATTC |
CGAGCGTCTTCTCTAAACCG |
58.4 |
105 |
|
Bra030779 |
AGAGAGCCACTGAACCCACA |
TTCCAACGTCCTTCTCCATC |
56.4 |
83 |
|
Bra034631 |
CAAAAGTGTTGCGGTTCTTG |
TCGACCACGGAAGTCTCAAT |
54.4 |
83 |
|
Bra017572 |
CCGTTGGCAGTCAGCGTTAT |
CAAAGCGTCGATTGCATACC |
56.4 |
85 |
|
Bra026682 |
TGCAGTTTGTGGACGAGGAG |
ATGTAGTAGTTTCAGTGTCCTCAAGC |
57.7 |
88 |
|
Bra026923 |
GCTTTTCCGGAGACAGCAT |
ACACAGCCAGAAGCTAAGCA |
55.3 |
96 |
|
Bra026924 |
TCTTTTTTGTTTGGTCTTAAAGTGT |
GCTAAACAATAATTAGAAAGGAAGGAA |
52.6 |
101 |
|
Bra026977 |
AGGATCATTTCAGATGGAGGAG |
ATGAGCGCATGAGTTCAGAG |
55.4 |
83 |
|
Bra026978 |
TTCCCGGAGACAGTATTGATTA |
TTTGTGGTTGTTGCTGTCTTAAC |
54.2 |
81 |
|
Bra026979 |
AAGAGGTGGCTAGTCCTGCTG |
GCATTGTTTCTTGACCAACG |
56.4 |
81 |
|
Bra027097 |
TTCTGGTCAAGAAACTCCGAT |
GCCTCTTCACGACTTCAGGT |
55.5 |
81 |
|
Bra027866 |
AGATGCAAACGCAAAAAAGC |
TCAGCGTCAAGAACAATCTCAT |
52.6 |
88 |
|
Bra036845 |
TCCCGTGAAGAAGAAGGATT |
CTGAGCTCGTAAGAACGCAC |
55.4 |
90 |
|
Bra037123 |
TCAGGCTTACGGCATAGGAA |
TCAACAGGTCGAGCACGTC |
56.3 |
100 |
|
Bra037139 |
CTTCCTTGTCTACCTTCTATCGTTT |
TCAGATTATAAGATTTTAGGAGACCTCT |
55.7 |
93 |
|
Bra002495 |
GGTGTGTGCAGGTGAAACCA |
GAGGGAACCGAAAAAATGATATAA |
55.1 |
99 |
|
Bra015597 |
TGGTTGAACTTTAATATACGGCTG |
GCTGAGCCCAAAGTGAGTGA |
55.9 |
91 |
- Regarding the information on the statistics used for the selection of inbred lines, I appreciate the answer of the authors. I suggest that the table with the statistics to be included as supplementary data.
Response:These data were added in supplementary data named as ’TableS6 The statistics of sensitive and resistant lines to DM and BR’.
- On page 17, line 517, point 4.1 of material and methods, I suggest the inclusion of the information provided by the authors on the cover letter addressing the review suggestions, more precisely, on points 19, 24 and 25.
Response:These pathogens of BR and DM were isolated from Chinese cabbage and stored in Chinese cabbage lab of Northeast Agricultural University. BR were incubated in LB for 30h and then spray on leaves with 107-108 Cfu/mL, however DM were directly wash by distilled water and spraying in vivo with 108 Cfu/mL with distilled water as control. The inbred lines of Chinese cabbage grew in greenhouse until the inoculation , and then fouth and fifth leaf were sprayed with BR and DM separately until outbursts.
The whole area of the leave and also the multiple yellowing and necrotizing lesions were measured by Multi-purpose leaf area meter LA-S(Wseen ). The specific value of these two area were calculated.
The statistics were shown as below
|
No. |
Inbred line A24 |
Inbred line A95 |
Inbred line A96 |
Inbred line A167 |
|||||||||||
|
Whole area (Mean) |
Lesion areas (Mean) |
Rate (%) |
Whole area (Mean) |
Lesion areas (Mean) |
Rate (%) |
Whole area (Mean) |
Lesion areas (Mean) |
Rate (%) |
Whole area (Mean) |
Lesion areas (Mean) |
Rate (%) |
||||
|
1-1 |
86 |
66 |
76.7% |
139 |
27 |
19.4% |
65 |
48 |
73.8% |
91 |
18 |
19.8% |
|||
|
1-2 |
92 |
72 |
78.3% |
89 |
17 |
19.1% |
107 |
86 |
80.4% |
117 |
20 |
17.1% |
|||
|
2-1 |
108 |
83 |
76.9% |
132 |
26 |
19.7% |
92 |
73 |
79.3% |
66 |
13 |
19.7% |
|||
|
2-2 |
105 |
86 |
81.9% |
140 |
29 |
20.7% |
118 |
88 |
74.6% |
68 |
11 |
16.2% |
|||
|
3-1 |
96 |
75 |
78.1% |
118 |
23 |
19.5% |
76 |
58 |
76.3% |
126 |
21 |
16.7% |
|||
|
3-2 |
90 |
71 |
78.9% |
106 |
21 |
19.8% |
128 |
96 |
75% |
123 |
23 |
18.7% |
|||
TableS6 The statistics of sensitive and resistant lines to DM and BR
- On point 4.5 of material and methods, I suggest more information such as, if the presence of gDNA was tested in crude RNA, how the efficiency of primers was determined, and what formula was used to determine the expression profile (Livak Hellmans, or other).
- Response:The primers were firstly amplified by the crude RNA to exam the solubility curve with one single peak and amplification curve with a obious ‘S’ type curve as preliminary experiments, and then the primers were used in the expression profile. Analysis of Relative Gene Expression Data Using Real-Time Quantitative PCR and the 2-△△CT Method(Livak Hellmans).
- Finally, I suggest that a small conclusion should added to the end of the discussion, in order to close and summarize the results of the good work presented.
Response:There was a short conclusion at the end of last paragraph in ‘Instruction’, show as below.
CC–NBS–LRR proteins in Chinese cabbage have not been explored, especially their roles in defense against bacterial and fungal infection. In our study, the CC–NBS–LRR genes of Chinese cabbage were identified and analyzed in a genome-wide range. A total of 40 genes encoding CC–NBS–LRR members were identified and further analyzed in terms of gene length and structure and chromosome location and distribution. The PI and conserved domains of CC-NBS-LRR proteins were also analyzed. A phylogenetic tree of CC–NBS–LRR genes in Brassica rapa, Arabidopsis thaliana, and Oryza sativa was built and used in exploring the homologous relationship between monocotyledons and dicotyledons. The expression profiles of BrCC–NBS–LRR genes were detected by infection with DM and BR. To further explore the roles of BrCC-NBS-LRR genes involved in pathogenesis-related defense, sensitive and insensitive inbred lines of the Chinese cabbage were infected by Hyaloperonospora parasitica and Xanthomonas campestris. campestris dowson for the purpose of drawing the expression profiles and detecting the differential expression of BrCC–NBS–LRR genes in our study. Our study provides information on BrCC–NBS–LRR genes. This information is useful to the further investigation of the gene functions and mechanisms of CC–NBS–LRR genes in Chinese cabbage.
